# Offshore CO$_2$ Capture and Utilization Using Floating Wind/PV Systems: Site Assessment and Efficiency Analysis in the Mediterranean

Douglas Keller, Jr. [1,*], Vishal Somanna [1], Philippe Drobinski [1] and Cédric Tard [2]

1   LMD/IPSL, École Polytechnique, Institut Polytechnique de Paris, ENS, PSL Research University, Sorbonne Université, CNRS, 91120 Palaiseau, France
2   LCM, École Polytechnique, Institut Polytechnique de Paris, CNRS, 91120 Palaiseau, France
*   Correspondence: dg.kllr.jr@gmail.com

**Abstract:** A methanol island, powered by solar or wind energy, indirectly captures atmospheric CO$_2$ through the ocean and combines it with hydrogen gas to produce a synthetic fuel. The island components include a carbon dioxide extractor, a desalinator, an electrolyzer, and a carbon dioxide-hydrogen reactor to complete this process. In this study, the optimal locations to place such a device in the Mediterranean Sea were determined, based on three main constraints: power availability, environmental risk, and methanol production capability. The island was numerically simulated with a purpose built python package **pyseafuel** . Data from 20 years of ocean and atmospheric simulation data were used to "force" the simulated methanol island. The optimal locations were found to strongly depend on the power availability constraint, with most optimal locations providing the most solar and/or wind power, due to the limited effect the ocean surface variability had on the power requirements of methanol island. Within this context, optimal locations were found to be the Alboran, Cretan, and Levantine Sea due to the availability of insolation for the Alboran and Levantine Sea and availability of wind power for the Cretan Sea. These locations were also not co-located with areas with larger maximum significant wave heights, thereby avoiding areas with higher environmental risk. When we simulate the production at these locations, a 10 L s$^{-1}$ seawater inflow rate produced 494.21, 495.84, and 484.70 mL m$^{-2}$ of methanol over the course of a year, respectively. Island communities in these regions could benefit from the energy resource diversification and independence these systems could provide. However, the environmental impact of such systems is poorly understood and requires further investigation.

**Keywords:** solar; wind; Mediterranean; methanol; simulation model; carbon capture

## 1. Introduction

Carbon dioxide (CO$_2$) levels in the atmosphere continue to rise, despite efforts to limit sources of anthropogenic emissions, such as through the Paris Agreement to limit global warming to under 2 °C [1]. In March 2022, the monthly global average atmospheric CO$_2$ was at a concentration of 418.28 ppm, almost 3 ppm more than the value in March 2021 (Ed Dlugokencky and Pieter Tans, NOAA/GML; gml.noaa.gov/ccgg/trends/; last accessed 13 June 2022).

To combat this continuing increase, carbon dioxide removal (CDR) is being extensively studied as a way to potentially reverse emissions enough to return warming trends to only a 1.5 °C global temperature increase [2,3]. CDR technologies include producing bioenergy with carbon capture and storage (BECCS), where crops are used to produce biofuels from captured atmospheric CO$_2$ [4], direct air carbon dioxide capture and storage (DACCS), which extracts CO$_2$ from flue or exhaust gas in industrial applications [5–7], and indirectly capture through the extraction of CO$_2$ from the ocean [8–12]. In the latter example, the decrease in aqueous CO$_2$ reduces its partial pressure in the ocean, leading to

the atmospheric $CO_2$ to equilibrate with the ocean and reinsert atmospheric $CO_2$ into the waters [13].

One of the most studied methods of indirect ocean capture is ocean liming [8,14]. This process shifts the ocean carbonate system (explained in more detail in Section 2.3) to be more basic, reducing the aqueous $CO_2$ and increasing the intake of atmospheric $CO_2$ through the mechanism mentioned above. However, lime production itself is a highly $CO_2$ emitting process, reducing the overall benefit of ocean liming [15]. Another approach involves mimicking the ocean's biological pump by mechanically pumping acidic surface layer waters below to deep less acidic waters, inducing vertical mixing to effectively increase the surface layer's pH and encourage atmospheric $CO_2$ uptake [16]. Further approaches directly accelerate the biological pump by fertilizing phytoplankton growth with iron, again reducing the aqueous $CO_2$ near the surface and encouraging atmospheric $CO_2$ uptake [17–19].

An alternative method to those described above involves the extraction of ocean $CO_2$ via an electrochemically induced pH swift [10–13,20–30]. This method shifts a flow of seawater more acidic, causing the flow to release gaseous $CO_2$ that can then be collected. After collection, the $CO_2$ can be combined with hydrogen to produce a synthetic fuel [31–34], which can then be used in place of fossil fuels, completing the anthropogenic carbon cycle, or stored, removing the carbon from the Earth's carbon cycle. One such device that incorporates this process is the renewable energy powered methonal-producing island [32] (referred to as a methanol island from now on), shown in Figure 1. The device is composed of floating photovoltaic panels [35] used to power the extraction of carbon and production of the synthetic fuel. More explicitly, the device operates by extracting the atmospheric $CO_2$ dissolved in the seawater with an electrochemical extraction module and combines it with hydrogen gas, sourced from desalinated water that is separated into hydrogen and oxygen gas. The carbon dioxide and hydrogen gasses react in a reactor, producing methanol [33].

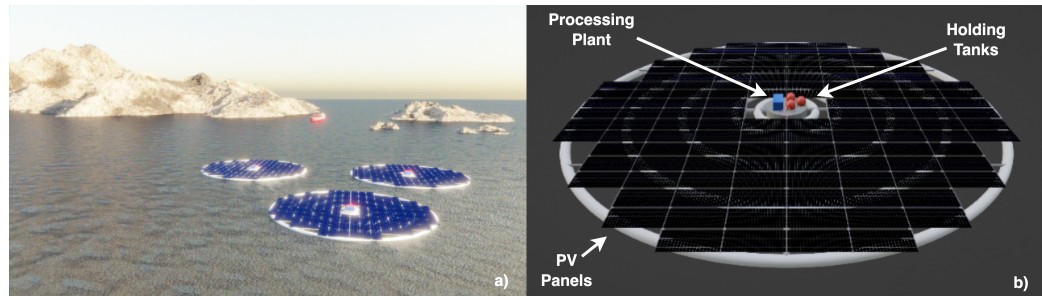

**Figure 1.** Artistic rendering of solar methanol island. (**a**) shows the device in a potential deployment scheme, serviced by a tanker ship. (**b**) shows a close-up of the device with its different main components: the floating structure and solar panels, the processing plant, and the holding tanks for the produced methanol. Further detail of the device's processes are given in Section 2.3.

In our study, we are interested in determining the optimal locations to put such a device in the Mediterranean Sea. It was marked as a potential location in Patterson et al., 2019 [32] for solar methanol islands, due to its high insolation. Other studies have found the Mediterranean to be a hotspot for solar and wind power [36,37], with strong persistent winds that can also become a hazard to floating structures [38]. Close to 7.6% of the world's population lives in the region [36], with a steadily increasing oil and gas consumption. Additionally, these devices could be used to improve the independence and self sustainability of remote and island communities by providing a means of local fuel production. There are over 191 islands in the Mediterranean, with 7 that have a significant population size (Corsica, Crete, Cyprus, Majorca, Malta, Sardinia, Sicily). This makes the overall region a prime location to study the placement of an alternative, renewable fuel source that can be locally produced.

To determine the optimal locations, we numerically simulate a methanol island and its production over 20 years, from 1993 to 2013. Atmospheric and oceanic model data

from the RegIPSL model are used to drive the methanol island simulation. In order to incorporate the large volume of the atmospheric and oceanic data [39] to properly examine the spatial variability of methanol production, a python package called **pyseafuel** was developed; a capability not available through other chemical process simulation software, as they are plant centric (COCO, https://www.cocosimulator.org/index.html; last accessed 10 November 2022, DWSIM [40], nor Aspen HYSYS, https://www.aspentech.com/en/products/engineering/aspen-hysys; last accessed 10 November 2022).

Our criteria for optimal locations is comprised of three main constraints:

- Available power,
- Environmental risk,
- Available methanol production.

The power constraint will be analyzed in terms of available wind and solar energy in the region. Other energy extraction methods exist, but are not as mature as wind or solar [37]. Environmental risk is assessed through the maximum wave heights found in different parts of the region. The available methanol production is analyzed in terms of the volumetric flow rate per Watt and per required area of power generation. Then, the methanol production will be compared to the energy consumption of two island communities to determine its potential use for these communities.

The organization of the paper is as follows: Section 2 outlines the modelling of the methanol island through the **pyseafuel** python package and covers the atmospheric and ocean model data used to drive said modelling. Section 3.1 presents the power availability of the region for the two power resources. Section 3.2 presents the maximum wave heights calculated for the region. Section 3.3 covers the simulated methanol production and its results. Section 3.4 presents the optimal locations. Section 3.5 discusses the potential use of the device in an island community setting. Concluding comments are presented in Section 4.

## 2. Materials and Methods

To numerically simulate the methanol island in the Mediterranean, data from the Earth system modelling from the years of 1993 to 2013 are utilized. The variables utilized and their flow paths to determine available power and available methanol production are provided in Figure 2. To determine the solar and wind power available to the methanol island, 10 m wind speeds and shortwave downward radiation from an atmospheric model are used in conjunction with two python packages, **windpowerlib** and **pvlib**. The environmental risk is considered through the modelled maximum wave heights, which are calculated from the 10 m wind speed. The sea surface temperature and salinity from an oceanic model are provided to a purpose built python package for this study, **pyseafuel**, to model methanol production. Figure 2 shows the data flow and structure of the analysis in the study.

The atmospheric model data used in this study are the outputs of a RegIPSL simulation, the regional climate model of IPSL [41]. This run used the coupling of the Weather Research and Forecasting Model (WRF) [42], a fully compressible and nonhydrostatic atmospheric model, and the ORCHIDEE Land Surface Model [43], which models the energy and water balance and vegetation of the terrestrial biosphere. The run was a hind-cast simulation (ERA Interim downscaling), performed at 20 km resolution, spanning the period of 1979 to 2016, within the Med-CORDEX framework [44]. The domain is shown in Figure 3a. In addition to being used to calculate the available solar and wind power and maximum wave heights, the 10 m horizontal wind components and shortwave radiation are used to force the ocean modelling described below. The 2 m temperature, 2 m specific humidity, longwave radiation, precipitation, and snowfall from the atmospheric model data were also used to force the ocean modelling.

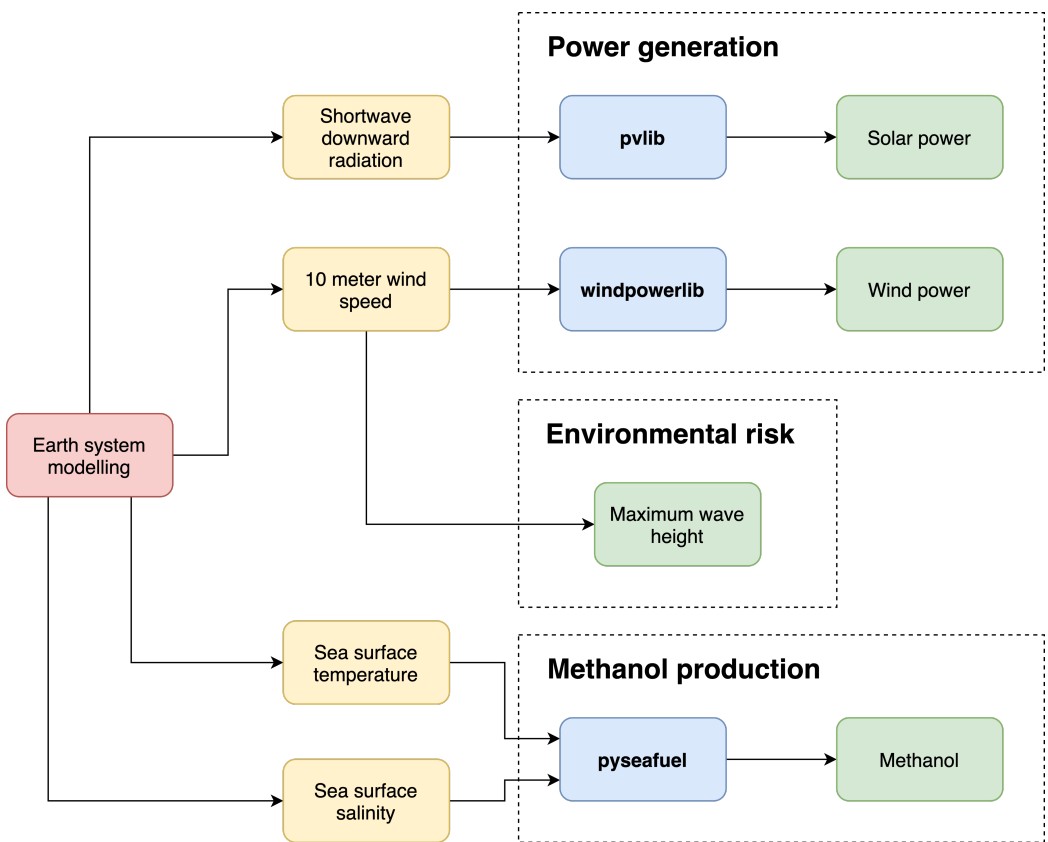

**Figure 2.** Earth system modelling provides shortwave downward radiation, 10 m wind speed and direction, sea surface temperature, and sea surface salinity. The shortwave downward radiation and 10 m wind speed are used to determine the power availability in the region. The 10 m wind speed is used to determine maximum wave height. The sea surface temperature and salinity are used to determine the methanol production.

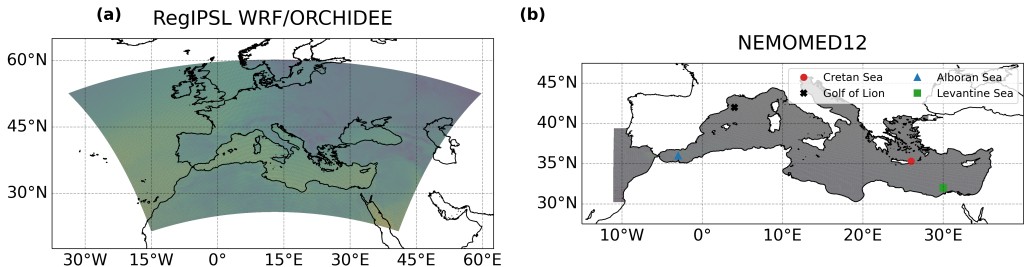

**Figure 3.** Domains of the WRF/ORCHIDEE simulation (**a**), and NEMO simulations (**b**). Both were run within the RegIPSL regional model.

The **N**ucleus for **E**uropean **M**odelling of the **O**cean (NEMO) ocean model (https://www.nemo-ocean.eu/; last accessed: 13 June 2022) is used to simulate the Mediterranean Sea in one year runs for 20 years, from 1 July 1993 to 30 June 2013. The hydrostatic, incompressible ocean model is run in bulk configuration to calculate the surface fluxes, utilizing the aforementioned atmospheric variables. In addition to using the bulk configuration, the NEMO model (v3.6) is also run in the NEMOMED12 configuration. NEMOMED12 is described, with boundary conditions, in [45–48]; a brief description follows: the domain covers the Mediterranean Sea and a portion of the Atlantic Ocean (see Figure 3b). The latter buffer zone is used to represent the exchanges between the two bodies of water at the Strait of Gibraltar, and its sea surface height (SSH) fields are restored towards the ORAS4 global ocean reanalysis [49]. The 3-D temperature and salinity fields of the buffer zone are restored

towards the MEDRYS reanalysis [46]. The Black Sea, runoff of 33 major rivers, and coastal runoff are represented by climatological data from [50]. The initial conditions for each one year run were pulled from the MEDRYS reanalysis [46]. The sea surface temperature and salinity from this model data are used to simulate the power required for the methanol production process.

*2.1. Power Generation*

2.1.1. Solar

The effective irradiance and solar panel power are calculated using the **pvlib** python library [51] (source code available at https://github.com/pvlib/pvlib-python; last accessed 11 November 2022). The global horizontal irradiance (GHI) is provided with the shortwave downwelling radiation from the WRF/ORCHIDEE model data. The direct normal irradiance (DNI) is determined using the DISC model [52] with the NREL implementation. The diffuse horizontal irradiance (DHI) is then calculated from GHI and DNI (https://pvpmc.sandia.gov/modeling-steps/1-weather-design-inputs/irradiance-and-insolation-2/global-horizontal-irradiance/; last accessed 13 June 2022):

$$DHI = GHI - DNI \cos \theta_z, \tag{1}$$

where $\theta_z$ is the zenith angle of the sun. The effective irradiance is then determined with the SAPM model from Sandia National Laboratory [53] and assuming an isotropic atmosphere. The SAPM model is also used to calculate the power generation of an arbitrary commercial solar panel, a 2009 SunPower 128-Cell Module, the data of which is provided in **pvlib** [51].

2.1.2. Wind

As stated above, the 10 m horizontal wind components are provided by the WRF/ORCHIDEE model run. However, to simulate the output of an offshore wind turbine, the London Array wind farm will be used as a template, which operates 175 Siemens Gamesa SWT-3.6-120 turbines. These turbines have a hub height of 87 m. Therefore, the wind speed at 87 m is estimated with the empirical power law [54]:

$$U_e = U_r \left( \frac{z}{z_r} \right)^{\alpha}, \tag{2}$$

where $U_e$ is the estimated wind speed at height $z$. $U_r$ is the reference wind speed at reference height $z_r$, in this case at 10 m. $\alpha$ is the empirical coefficient unique to different locations and surface types. For this study, we use $\alpha = 0.11$ [55] for the open sea. From the estimated 87 m wind speed, the estimated power production of a SWT-3.6-120 turbine can be calculated.

The estimated power production from a Siemens Gamesa SWT-3.6-120 turbine is calculated with the **windpowerlib** python package [56] (source code available at https://github.com/wind-python/windpowerlib; last accessed 11 November 2022), which provides the power curve for the turbine (see Figure 4). A noteworthy feature of the turbine power curve is the power cutoff if too high of wind speeds are encountered. This cutoff is common to many commercially available turbines to prevent damage from stronger winds. This also limits the productivity of the areas with the highest wind speeds. This turbine has a rotor diameter of 120 m, which is used to calculate the power generation in terms of power per swept area of the turbine, allowing it to be compared to the solar panel power generation.

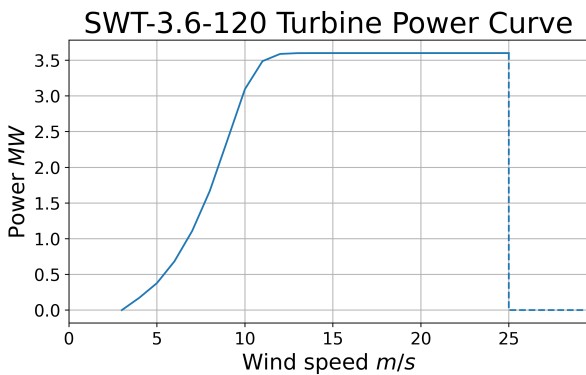

**Figure 4.** Power curve for the SWT-3.6-120 turbine. Provided by the **windpowerlib** python library [56].

### 2.2. Environmental Risk

#### 2.2.1. Max Wave Height

The maximum wave height is calculated with the following empirical relationship [57], which assumes a fully developed sea:

$$H_{max} = 2\left(\frac{U}{12.5}\right)^2, \tag{3}$$

where $H_{max}$ is the maximum wave height, and $U$ is the 10 m wind speed in knots. Accurately modelling wave heights is a much more complex topic than Equation (3) alludes to, requiring information such as fetch, sea depth, wind direction, and basin geometry and is not calculated in the NEMO ocean model utilized in this study. Therefore, we use Equation (3) to determine potential areas of concern for this study.

### 2.3. Methanol Production—*pyseafuel*

A python package, **pyseafuel** (source code available at https://gitlab.in2p3.fr/energy4climate/public/pyseafuel; last accessed 11 November 2022), was specifically made by the authors to numerically simulate the methanol island for this study. This Section 2.3 describes the structure and theory used in the package. Four subprocesses make up methanol island, as seen in Figure 5, and are simulated by **pyseafuel**: degassing, desalinating, electrolyzing, and reacting. Seawater is fed into both a degasser and desalinator in separate flows. The degasser extracts the $CO_2$ from the seawater and passes it along to the reactor. The degassed seawater is expelled at this point. Meanwhile, the seawater fed into the desalinator is purified (expelling the brine back to the ocean) and passed along to an electrolyzer. The electrolyzer then separates the purified water into hydrogen gas ($H_2$) and oxygen gas ($O_2$). The oxygen is expelled to the atmosphere. The hydrogen is passed to the reactor alongside the carbon dioxide, where they react to form methanol ($CH_3OH$). Of the four subprocesses, only the reactor subprocess is exothermic and does not require additional power except that required to keep it at its operating temperature and pressure. The other three subprocesses all require a supply of power to operate. The subprocesses are described separately below.

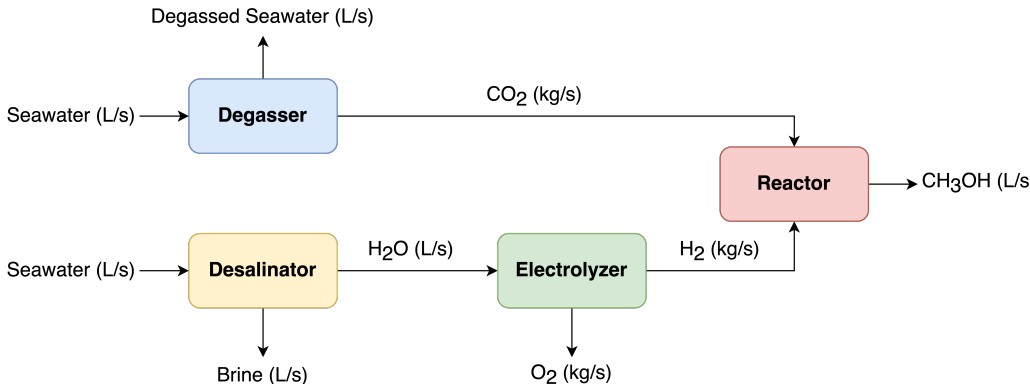

**Figure 5.** Layout of a methanol island simulated by **pyseafuel**. Two arms are fed seawater, one through the degasser to extract $CO_2$ and one through the desalinator and electrolyzer to produce $H_2$. The $CO_2$ and $H_2$ is then combined in the reactor to produce $CH_3OH$, methanol. Waste brine and $O_2$ are produced as part of the process.

### 2.3.1. Degassing

**pyseafuel** calculates the degassing subprocess using experimental results from Eisaman et al., 2012 [29]. $CO_2$ dissolved into the ocean immediately reacts with ocean carbon buffer system, represented by the following equilibrium equations [58]:

$$CO_2(g) = CO_2(aq), \tag{4}$$

$$CO_2(aq) + H_2O = H_2CO_3, \tag{5}$$

$$H_2CO_3 = H^+ + HCO_3^-, \tag{6}$$

$$HCO_3^- = H^+ + CO_3^{2-}. \tag{7}$$

As the two species are difficult to distinguish from one another, aqueous carbon dioxide ($CO_2(aq)$) and carbonic acid ($H_2CO_3$), are typically grouped together as $CO_2^*$. These equilibrium equations rely on the available hydrogen ions ($H^+$), as observed in Equations (6) and (7), the equations for carbonic acid and carbonate ($CO_3^{2-}$). The carbon dioxide extraction process implemented currently in **pyseafuel** is based upon the method employed by Eisaman et al., 2012 [29]. This method makes use of the dependence of the buffer system on the freely available hydrogen ions. If the buffer system is shifted to a very acidic solution (pH < 4.5), almost all of the carbon shifts to exist in the form of $CO_2^*$, as observed in Figure 6, and readily off-gasses $CO_2(g)$. The off-gas $CO_2(g)$ can then be captured and utilized.

Eisaman et al., 2012 [29] performs this pH shift by splitting the flow into two separate streams, acidifying one stream and basifying the other by using a series of bipolar membranes with an applied potential difference. The off-gassed $CO_2(g)$ is then collected with a membrane contactor from the acidified stream. The two streams are then recombined, returning the overall flow to a neutral pH, which is then discarded. This process is simulated in **pyseafuel** by fitting quadratic curves to the experimental data provided in the Supplementary Information of Eisaman et al., 2012 [29]. The fitted curves for both the energy consumption per mole of extracted $CO_2(g)$ versus pH and pH versus the ratio of the out-flowing $CO_2(g)$ (at STP L $min^{-1}$) and in-flowing seawater (L $min^{-1}$) data are shown in Figure 7. The power requirement and $CO_2(g)$ extraction of this subprocess are then estimated from these curves.

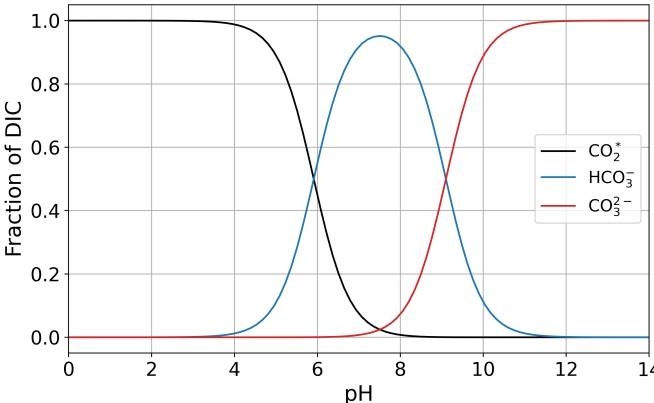

**Figure 6.** The ocean carbon buffer system. The equilibriums and dominant species were calculated using Equation (4) through (7). Dissolved inorganic carbon (DIC) is the sum of the three species.

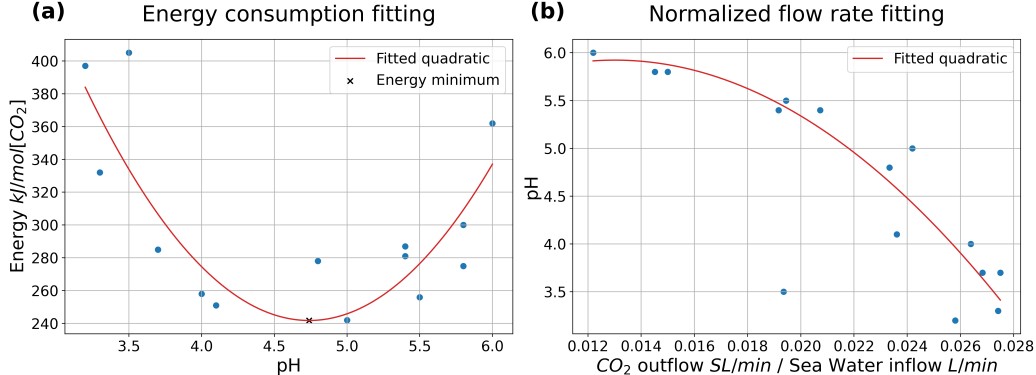

**Figure 7.** The fitted quadratic curves for the energy consumption per mole of extracted $CO_2$ depending on pH (**a**), and for pH versus the ratio of outflow $CO_2$ over inflow seawater (**b**). The fitted equation for (**a**) is $58.29x^2 - 524.44x + 1423.21 = 0$ where $x = pH$, and for (**b**) is $-1195.47x^2 + 311.11x + 3.90 - pH = 0$ where $x =$ the $CO_2$ outflow / seawater inflow ratio.

### 2.3.2. Desalination

The desalination subprocess is calculated with **pyseafuel** by assuming the process is performed with electrodialysis. Electrodialysis desalinates seawater by applying a potential difference over a series of cation and anion exchange membranes (CEMs and AEMs; inclusively called ion exchange membranes, IEMs) [59]. These series of IEMs separate the incoming saline flow (feed flow) into a more saline brine flow and a nearly purified fresh water flow, depending on the desired separation. Keeping the power constraint in mind to determine optimal locations for the device, the equation for energy consumption, *EC* (J L$^{-1}$), for this subprocess for a single stage setup is (assuming perfect exclusion of the correct ions at the IEMs) [59]:

$$EC = 2RT_f(c_f - c_p) \ln \frac{c_b}{c_p},\qquad(8)$$

where *R* is the ideal gas constant (taken as 8.3145 J mol$^{-1}$ K$^{-1}$), $T_f$ is the temperature of the feed flow in Kelvin, and $c_f$, $c_p$, and $c_b$ are the salinity concentrations of the feed, product, and brine flows (mol L$^{-1}$). For a multistage process with *n* stages, the following equation can derived (using the relevant information from Wang et al., 2020 [59]:

$$EC = 2RT_f\zeta_n \ln \left( \frac{1}{(1-\zeta_n)(1-R_w^{1/n})} - \frac{R_w^{1/n}}{1-R_w^{1/n}} \right) c_f \sum_{i=1}^{n}(1-\zeta_n)^{i-1},\qquad(9)$$

where $\zeta_n = 1 - (1 - \zeta)^{1/n}$ is the salt removal percentage per stage $n$, with $\zeta$ as the overall salt removal percentage of the subprocess. $R_w$ is the ratio of the fresh water outflow over the saline feed flow.

### 2.3.3. Electrolysis

The electrolysis subprocess is calculated in **pyseafuel** using the simplified model for proton (or polymer) exchange membrane (PEM) electrolysis, derived by Shen et al., 2011 [60]:

$$P_e = IV = K(V - IZ - E_0)^2 + I^2 Z, \tag{10}$$

where $P_e$ is the subprocess power consumption per stack area (W cm$^{-2}$), $I$ is the operating current per stack area (A cm$^{-2}$), $V$ is the operating voltage (V), $K$ is the power coefficient ($\Omega^{-1}$ cm$^{-2}$), $Z$ is the internal resistance of the electrolyzing cell per stack area ($\Omega$ cm$^{-2}$), and $E_0$ is the cell reversible potential of electrolysis for the separation of water into hydrogen and oxygen gas (V). PEM electrolysis works by applying a voltage across a membrane in contact with a pure water flow. The pure water dissociates into H$^+$ ions and O$_2$ from the applied potential and the PEM then only permits the H$^+$ ions to cross, whereby the ions then combine with electrons to form H$_2$. The value for $Z$ is measured for a given experimental cell, then $K$ can be determined from fitting Equation (10) to the experimentally measured $I$ and $V$. In implementation, $I$ is determined by the desired outflow of H$_2$ (and therefore required inflow of pure water) and then Equation (10) is solved for in terms of $V$, given $K$, $Z$, and $E_0$ are known. $P_e$ can then be computed.

### 2.3.4. Reactor

The reactor subprocess is simulated in **pyseafuel** using the plug flow reactor model described by Terreni et al., 2020 [33] (Figure 8). The following equations describe the model:

$$\frac{d\dot{N}_i(x)}{dx} = \rho_{cat} A_{tubes} R_i(\mathbf{p}(x)), \tag{11}$$

$$\mathbf{p} = \frac{P}{P_0} \frac{\dot{\mathbf{N}}}{\sum_j \dot{N}_j}, \tag{12}$$

where $\rho_{cat}$ is the density of the catalyst in the plug flow tubes, $A_{tubes}$ is the cross sectional area of all the tubes (number of tubes multiplied by $a_{tubes}$; see Figure 8), and $\mathbf{p}$ is the array of reduced partial pressures, with $P$ as the reactor operating pressure (bars) and $P_0$ as the reference pressure (taken as 1.01325 bars). The $i$ and $j$ are the indexes for the following arrays:

$$\dot{\mathbf{N}} = (\dot{N}_{CO}, \dot{N}_{CO_2}, \dot{N}_{H_2}, \dot{N}_{H_2O}, \dot{N}_{CH_3OH}), \tag{13}$$

$$\mathbf{R} = (R_{CO}, R_{CO_2}, R_{H_2}, R_{H_2O}, R_{CH_3OH}) = (-r_1 + r_2, -r_2 - r_3, -2r_1 - r_2 - 3r_3, r_2 + r_3, r_1 + r_3). \tag{14}$$

where $\dot{\mathbf{N}}$ is the array of molar flows of the labeled species and $\mathbf{R}$ is the array of composite reaction rates for the labeled species, with the individual subcomponents described in the following:

$$r_1 = k_1 K_{CO} \left[ P_{CO} P_{H_2}^{3/2} - \frac{P_{CH_3OH}}{P_{H_2}^{1/2} K_1^{eq}} \right] / D, \tag{15}$$

$$r_2 = k_2 K_{CO_2} \left[ P_{CO_2} P_{H_2} - \frac{P_{H_2O} P_{CO}}{K_2^{eq}} \right] / D, \tag{16}$$

$$r_3 = k_3 K_{CO_2} \left[ P_{CO_2} P_{H_2}^{3/2} - \frac{P_{CH_3OH} P_{H_2O}}{P_{H_2}^{3/2} K_3^{eq}} \right] / D, \tag{17}$$

$$D = (1 + K_{CO}P_{CO} + K_{CO_2}P_{CO_2}) \left[ P_{H_2}^{1/2} + \frac{K_{H_2O}}{K_{H_2}^{1/2}} P_{H_2O} \right], \tag{18}$$

where $r_m$ are the reaction rates ($m = 1 \to 3$), $P_j$ are the partial pressures of the $j$ components, and $k_m$ and $K_*$ are the equilibrium coefficients, available in Terreni et al., 2020 [33]. In the reactor, the $H_2$ and the $CO_2$ interact with the catalyst to combine to form $CH_3OH$, however they can also combine to form carbon monoxide (CO) and water ($H_2O$) as part of the reverse water gas shift (RWGS) reaction [33]. The effectiveness of the conversion from the $CO_2$ to $CH_3OH$ depends strongly on the operating pressure, $P$, and temperature $T$ of the reactor, as well as on the ratio of the initial molar flows of $CO_2$ and $H_2$ ($r = \dot{N}_{H_2}^0 / \dot{N}_{CO_2}^0$). The dependence on $P$ and $T$ can be observed in Figure 9a, where the fractional conversion of initial $CO_2$ to $CH_3OH$, $\xi$ ($= \dot{N}_{CH_3OH} / \dot{N}_{CO_2}^0$), is larger with larger pressures and lower temperatures. $\xi$ is calculated using the equilibrium model provided in Terreni et al., 2020 [33] (see the article for details).

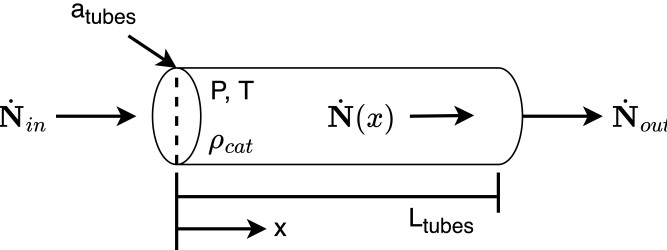

**Figure 8.** A single tube for a plug flow reactor. $a_{tubes}$ is the single tube area. $P$ and $T$ are the reactor operating pressure and temperature. $x$ is the coordinate along the length of the tube, which has a length of $L_{tubes}$. $\rho_{cat}$ is the density of the catalyst packed into the reactor tube. $\dot{N}(x)$ is the molar flow rate along the length of the tube, with subscripts *in* for the inflow and *out* for outflow.

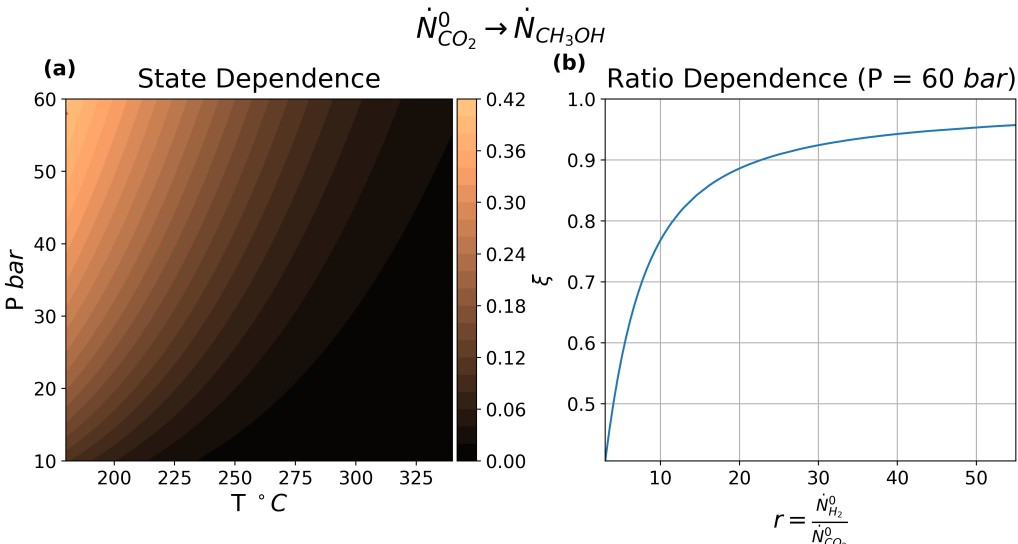

**Figure 9.** The conversion factor, $\xi$, or the percentage of moles of $CO_2$ transformed into $CH_3OH$. It was calculated using the equilibrium equations presented in Terreni et al., 2020 [33], with varying state conditions ($P$ and $T$) and molar ratios of $H_2$ and $CO_2$. The state dependence curve was calculated with a $H_2/CO_2$ molar ratio of 3, which is the stoichiometric ratio. Subplot (**a**) shows $\xi$'s dependency on pressure and temperature. (**b**) shows $\xi$'s dependency on the input ratio of $H_2$ and $CO_2$.

When looking at the initial ratio of molar flows in Figure 9b, "over-saturating" the reaction with $H_2$ improves $\xi$ (calculated with the same equilibrium model as before) significantly to a point, after of which there are diminishing returns with an increasing ratio. However, as will be demonstrated later in this study, the process of producing $H_2$

requires a significant amount of power, so a favorable *r* must be balanced with a reasonable power requirement.

## 3. Results and Discussion

### *3.1. Power Generation*

#### 3.1.1. Solar

The results for the 20 years are primarily presented in seasonal averages, arranged by the months of December, January, and February (DJF) for winter, March, April, and May (MAM) for spring, June, July, and August (JJA) for summer, and September, October, and November (SON) for fall. The results for the calculated and averaged effective irradiance are shown in Figure 10. The mean values over the sea range from 276.16 W m$^{-2}$ in the summer, to a low of 179.63 W m$^{-2}$ in the winter. As expected, the lowest values are found in winter, with the largest in summer. There is also a north/south variation, which is much more apparent during the winter and fall, due to the increased length of night during these seasons. The effective irradiance is essentially the available solar power that the solar panel can capture. As such, the solar panel power generation follows the same spatial variance as the effective irradiance, see Figure 11, just at reduced values. Mean values over the sea range from 118.56 W m$^{-2}$ in summer to 82.93 W m$^{-2}$ in winter, with intermediate values in the shoulder seasons. The resulting prime locations for solar power are along the north coast of Africa and the Levantine Sea, as these areas benefit from larger amounts of insolation during the fall and winter months.

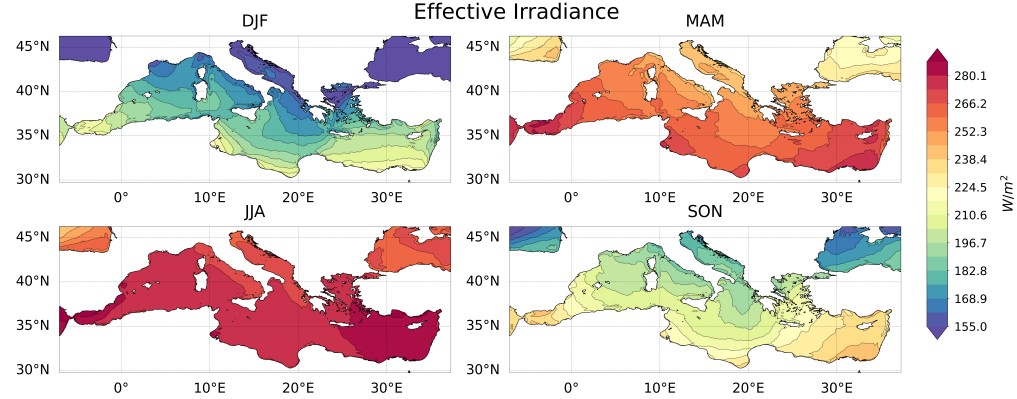

**Figure 10.** Seasonally averaged effective irradiance calculated from with the **pvlib** python package and WRF/ORCHIDEE shortwave radiation data.

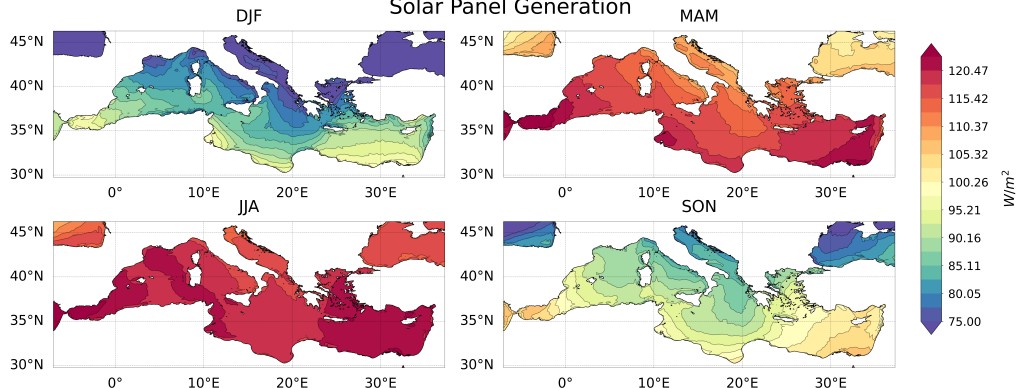

**Figure 11.** Same as Figure 10 but for the calculated solar panel power generation.

3.1.2. Wind

The seasonally averaged winds are shown in Figure 12. Two main regions are affected by strong winds in the Mediterranean: the Aegean Sea during the entire year but with the strongest winds in summer, the Etesians, where seasonal average speeds over 6 m/s can be found, and the Gulf of Lion during the winter, where it experiences the Mistral winds that are more frequent and stronger during the winter months [61]. The calculated power generation per swept area for the SWT-3.6-120 turbine is shown in Figure 13. It follows the same trends as the winds, however, only the two areas mentioned above with stronger, more persistent winds show any meaningful production, despite that wind speed seasonal averages over 3 m/s can be found over most of the Mediterranean basin over most of the year.

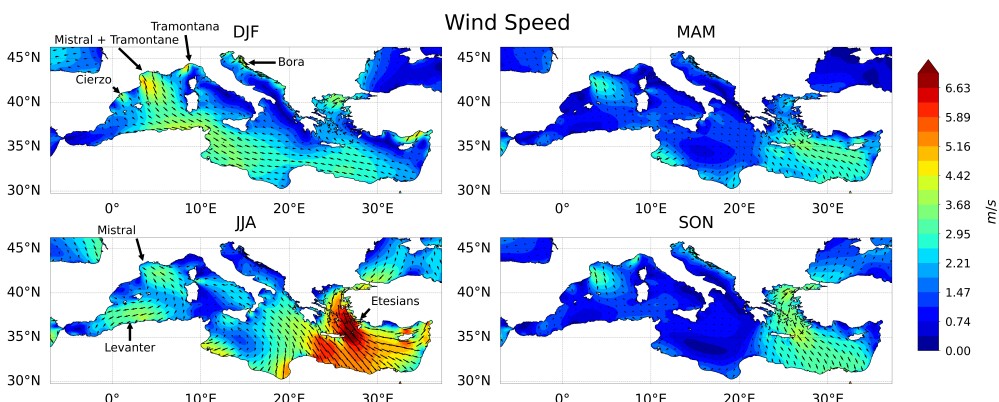

**Figure 12.** Seasonally averaged wind speeds and directions of the WRF/ORCHIDEE 10 m wind data. During the winter months, the Bora [62], Mistral and Tramontane [63,64], Tramontana [65], and Cierzo [66] winds can be observed. During the summer months, the summer Mistral [67], Levanter [65], and Etesian [68] winds can be observed.

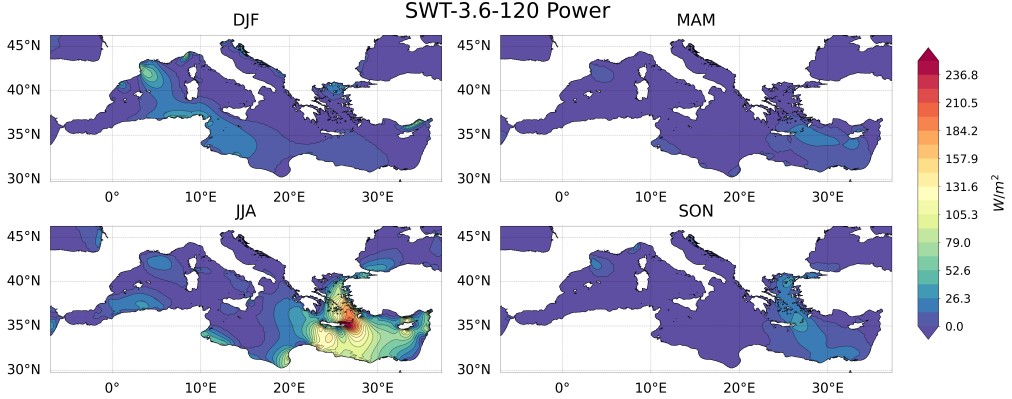

**Figure 13.** Seasonally averaged calculated power generation using a SWT-3.6-120 industrial turbine with height corrected wind speed data (with Equation (2)) from the WRF/ORCHIDEE 10 m wind speed data.

*3.2. Environmental Risk*

With regard to the potential hazard the methanol island structure can experience, the maximum wave height is just one ingredient needed to evaluate the risk to the structure. Much like accurately modelling ocean surface waves requires more variables than what is within the scope of this study, as discussed in Section 2.2.1, accurately measuring the risk to ocean installations that surface waves impose requires more information from the wave properties than just the maximum wave height. Variables such as the wave directions, periods, and height spectra are needed to estimate structure loading [32], as well as details

about the structure itself, for both floating solar panels [69] and offshore wind turbines [70]. However, the maximum wave height gives us preliminary guidance for areas to avoid when placing a methanol island.

The maximum wave heights over the 20 year period are shown in Figure 14. The largest wave heights are co-located with the strongest winds, as expected, given the wind dependency in Equation (3). Overall, there are larger values in the western Mediterranean basin than in the eastern basin. In particular, areas affected the most include (moving from west to east) the Gulf of Lion, the Balearic Sea and parts of the Algerian basin, the Gulf of Gabes off the coast of Tunisia, the Ionian Sea, and the Aegean Sea, with most of the peak values in these areas at around 2.5 m. Otherwise, the rest of the Mediterranean shows values under 1.75 m. These results appear to agree with the general spatial variability of the significant wave height, which is related to the maximum wave height ($H_{max} \approx 2H_S$, where $H_S$ is the significant wave height; [57]), presented in Galanis et al., 2011 [71], with the exception that the Aegean Sea is less of a hotspot in their study than in our results. With this information, the aforementioned sites should be avoided for methanol island placement.

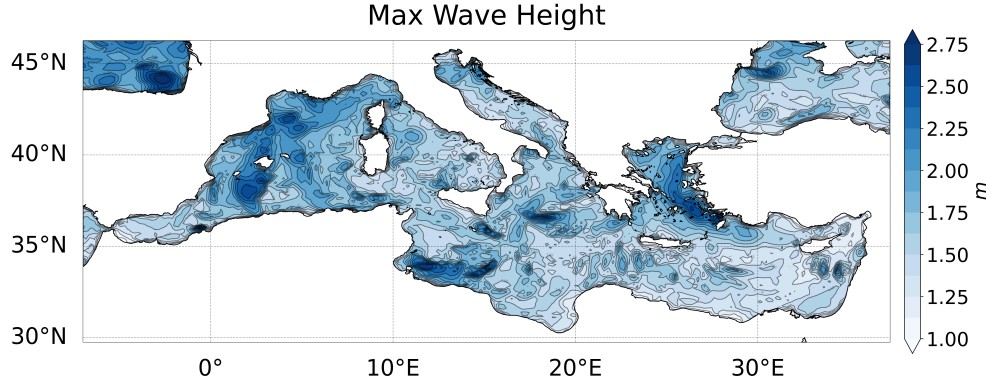

**Figure 14.** Calculated maximum wave heights using the 10 m wind speeds from the WRF/ORCHIDEE model outputs and Equation (3).

### 3.3. Methanol Production

The values used for the **pyseafuel** simulation are summarized in Table 1. The simulated methanol island is given a flow rate of 10 L s$^{-1}$ for the carbon production arm (the degasser), which is around 2/3 the flow rate of a French fire hydrant (1000 L min$^{-1}$). The hydrogen production arm (the desalinator and electrolyzer) is given a flow rate of 0.01 L s$^{-1}$, which is about a tenth of the flow rate from a European kitchen faucet (6 L min$^{-1}$). These are small flow rates and would be much larger in full scale application; however, our analysis will be confined to the "production efficiency" of the methanol island or the methanol production flow rate divided by the required power. This normalizes the analysis, making it applicable to studying the spatial variability of the production.

The flow rates for the two arms, 10 L s$^{-1}$ and 0.01 L s$^{-1}$, result in a $H_2$ to $CO_2$ mole ratio of 25.83. This results in a conversion factor for the plug flow reactor of 0.91, which is very similar to the equilibrium obtained conversion factor shown in Figure 9b.

The simulated island has three main power draws: from the degasser, the desalinator, and the electrolyzer. With the current configuration, only one depends on the spatial variability of the Mediterranean surface waters: the desalination subprocess, which depends on the sea surface temperature (Figure 15) and the sea surface salinity (Figure 16). However, for the flow rate simulated here, it is the lowest of the three power draws, ranging from 106.57 ± 78.6 W in the spring (mean ± standard deviation) to 109.17 ± 80.52 W in the summer, when compared to 2617.66 W consumed by the degassing subprocess and 2887.65 kW consumed by the electrolysis subprocess. Here, the electrolysis subprocess is by far the largest power draw, consuming multiple orders of magnitude larger than either the degassing or desalination subprocesses. Since this process produces the hydrogen for

the reactor, it is necessary to balance the benefit of an improved conversion factor, $\zeta$, from a higher hydrogen to carbon dioxide ratio (see Figure 9b) with the higher power cost of producing more hydrogen to optimize the efficiency of the overall methanol production.

Keeping in mind that only the desalination subprocess depends on spatial variability and is the smallest of the three subprocesses in terms of power consumption, the methanol production per power, in units of $\mu L\ day^{-1}\ W^{-1}$, varies very little over the Mediterranean basin. A mean value of $11.94\ \mu L\ day^{-1}\ W^{-1}$ is found over all the seasons, with the standard deviation changing from $\pm 0.00032\ \mu L\ day^{-1}\ W^{-1}$ in winter and spring to $\pm 0.00033\ \mu L\ day^{-1}\ W^{-1}$ in summer and fall.

**Table 1.** These are parameters and their values that are used to numerically simulate a methanol island with **pyseafuel**, separated by subprocess (Figure 5). The electrolyzer parameters are common values in industry [60,72]. *T* and *P* for the reactor are within the range of previously tested temperatures and pressures for the methanol production from carbon dioxide hydrogenation [73]. Note: around 2 V minimum is typically required to generate the necessary current densities for electrolysis to occur [74].

| Input and State Parameters | | | | |
|---|---|---|---|---|
| **Degasser** | | | | |
| Seawater inflow $(L\ s^{-1})$ | | | | |
| 10 | | | | |
| **Desalinator** | | | | |
| Seawater inflow $(L\ s^{-1})$ | $R_w$ | $n$ stages | $\zeta$ | |
| 0.01 | 0.5 | 1 | 0.99 | |
| **Electrolyzer** | | | | |
| $E_0$ (V) | $R\ (\Omega\ cm^{-2})$ | $K\ (\Omega^{-1}\ cm^{-2})$ | $A_{stack}\ (cm^2)$ | |
| 1.4 | 0.15 | 27.8 | 250 | |
| **Reactor** | | | | |
| $T\ (°C)$ | $P$ (bars) | $A_{tubes}\ (m^2)$ | $L_{tubes}\ (m)$ | $\rho_{cat}\ (kg\ m^{-3})$ |
| 180 | 60 | 3.14 | 3 | 1000 |

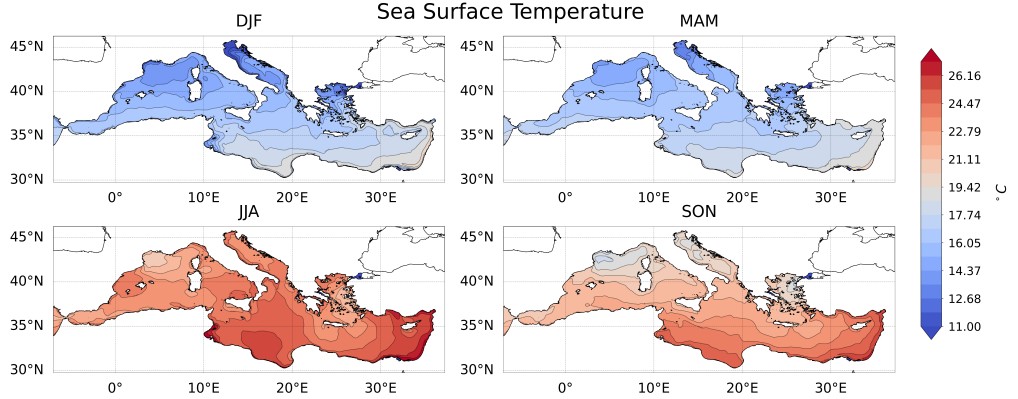

**Figure 15.** Seasonally averaged sea surface temperature from the NEMO simulation set.

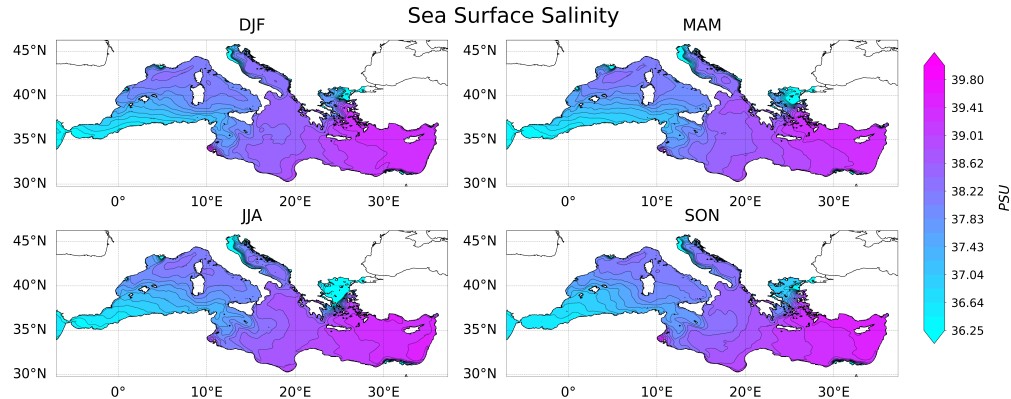

**Figure 16.** Seasonally averaged sea surface salinity from the NEMO simulation set.

To compare the methanol production powered by either solar power or wind power, the methanol production efficiency is multiplied by power generation in terms of W m$^{-2}$, resulting in a flow rate over area of power generation. For solar panels, this area is the area of panels needed for power generation, whereas for wind power, this area is the swept area of the turbine. Organizing the results in this manner allows us to compare areas that would need larger power generation installations versus areas that would need less for a given methanol flow rate. Figure 17 shows this flow rate per power generation area for a simulated island powered off of solar panels. Figure 18 shows the same but for an island powered off of wind energy.

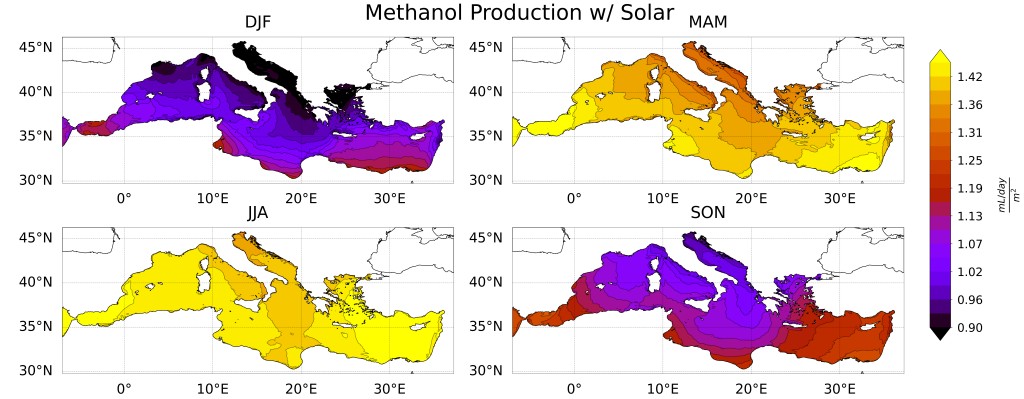

**Figure 17.** Methanol production powered by the solar panel power shown in Figure 11. Note the values are in milliliters per day per area.

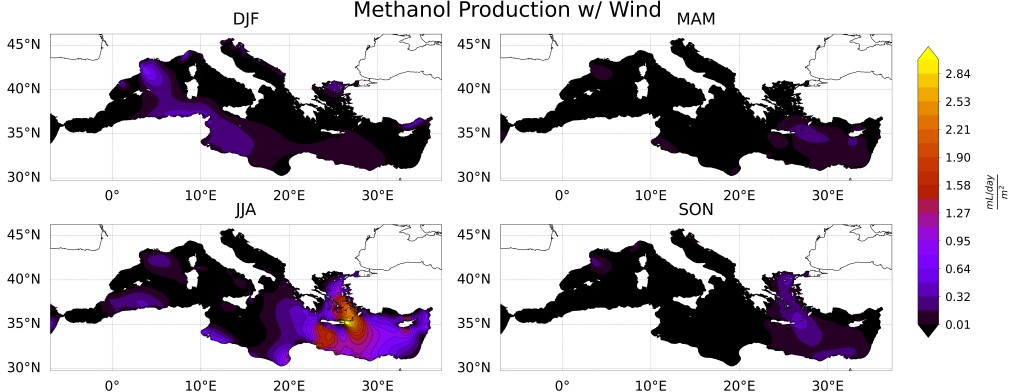

**Figure 18.** Same as Figure 17 but the production is powered with the turbine power from Figure 12.

As the methanol production per Watt is essentially constant over the entire Mediterranean basin, the results in Figures 17 and 18 essentially demonstrate the same results as Figures 11 and 12, respectively, as the power generation is just multiplied with an effectively constant coefficient. Larger production rates per power generation area are found in the southern part of the basin during the winter, spring, and fall for a solar panel powered methanol island, with high production rates over most of the basin during the summer. For the wind powered methanol island, higher production rates are found in the Gulf of Lion and Aegean Sea (as well as part of the Levatine), just as the wind power generation was higher in this regions as well, during their respective productive seasons.

Integrated Production

To look at how the methanol production changes over the course of the year with a finer temporal resolution; for example, locations have been selected to be examined in more detail. These locations are in the Alboran Sea, the Gulf of Lion, the Cretan Sea, and the Levantine Sea. Exact coordinates are given in Table 2 and the locations are shown in Figure 3b. These locations were selected as they were located in the solar power and wind energy supply hotspots pointed out in Section 3.1. The atmospheric and oceanic derived data were ensemble averaged over the 20 years into a single year. The results for solar panel and turbine power generation, and methanol production with solar power and wind energy are shown in Figure 19.

**Table 2.** Point locations examined in more detail due to their favorable solar and wind power generation.

| Location | Alboran Sea | Gulf of Lion | Cretan Sea | Levantine Sea |
|---|---|---|---|---|
| **Coordinates** | 36° N 3° W | 42° N 4° E | 35.30° N 26° E | 32° N 30° E |
| **Max Wave Height (m)** | 1.54 | 2.54 | 1.55 | 1.36 |
| **Integrated Methanol Production (mL/m$^2$)** | | | | |
| **Solar** | 494.21 | 445.52 | 465.10 | 484.70 |
| **Wind** | 20.80 | 219.60 | 457.29 | 152.85 |

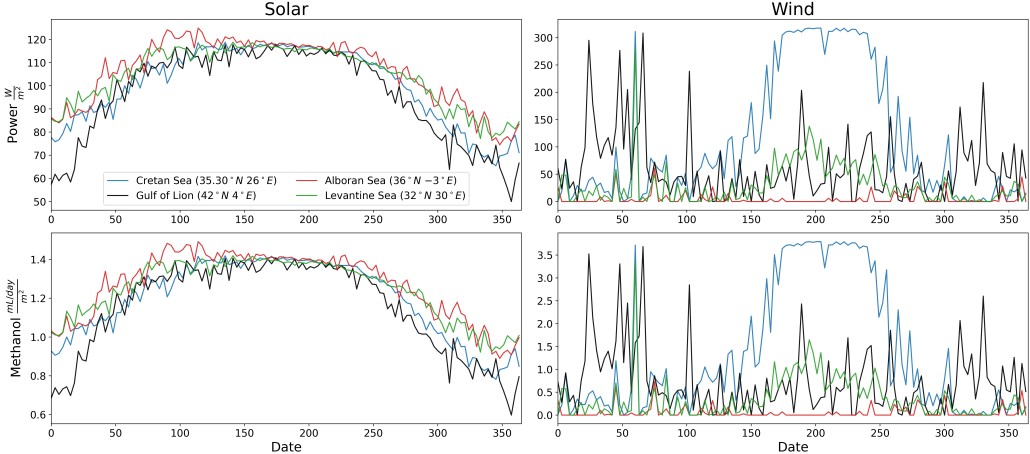

**Figure 19.** Ensemble time series for the point locations in Table 2. Solar panel and SWT-3.6-120 wind power generation are presented, as well as the corresponding methanol production from the two power resources. The solar panel power generation and methanol production time series are window averaged to make the temporal trends more visible, as production drops to zero every night.

What we observe from the figures is that the solar power generation is fairly consistent among the different locations, with maximum values approaching the 140 W m$^{-2}$ level in summer time and falling to around 60 W m$^{-2}$ in winter time (again in Figure 19, the values are window averaged to better show the trend and therefore hide some of the peak variability). The Gulf of Lion sees less power available from solar over the course of the

year, as it is further north than the other locations. However, both it and the Cretan Sea see more available power from wind energy, with values reaching 300 W m$^{-2}$ in both locations. In fact, the Cretan Sea location experiences a limit in generated power due to the SWT-3.6-120 power cutoff (Figure 4), and is limited to values under about 315 W m$^{-2}$. The main negative of the wind resource is its temporal variability with respect to solar power. While peak wind power values are around 300 W m$^{-2}$ compared to solar power's 140 W m$^{-2}$, values are often closer to zero during less productive times of the year. For example, the Cretan Sea location sees a peak during the summer and fall months, but has lower levels of power generation during the rest of the year, or off season. The Gulf of Lion shows a similar, yet less pointed trend with the on and off season timing reversed. The methanol production values follow the power curves closely for both the solar and wind energy, and thus reflect the same consistency/variability, just as they had in Figures 17 and 18. An important note however, is that the methanol production process operates as a quasi steady state process. It would be unable to handle extreme short time scale variations, potentially requiring the wind power to be smoothed before using it to power the island, whereas solar power may not.

To more accurately compare methanol production between the two power resources, the production was integrated over the year span, resulting in a single value per location. These values are presented in Table 2. Production amounts with solar power are similar across the different locations, with the Alboran and Levantine Sea locations producing the most. On the other hand, production amounts with wind energy are much more varied, following suit with the prior discussion. Only one location produced more methanol with wind than with solar power, the Cretan Sea location, which produced 7% more with wind. The runner up, the Gulf of Lion, produced 42% less with wind, even though it was the other wind energy hotspot in the Mediterranean.

This same procedure was conducted for all the spatial points in the Mediterranean and is presented in Figure 20, including the integrated production for solar, wind, and the difference between the two. The production based on solar power features the highest levels along the north coast of Africa, specifically in the Alboran and Levantine Sea. The production based on wind power features the largest levels in the Gulf of Lion, Aegean Sea, and parts of the Levantine Sea. Unsurprisingly, the simulated island produces more powered by solar power everywhere except for around the eastern side of the island of Crete, where wind produces slightly more.

### 3.4. Optimal Locations

Pooling together the results from Sections 3.2 and 3.3 allows us to label optimal locations for both solar and wind powered methanol producing islands.

As previously stated in Section 3.2, more information is needed to determine the limiting factors more accurately when it comes to the effect of environmental hazards. However, information on the limiting significant wave height for certain installations procedures for offshore wind farms are used to estimate sensitive areas to wave height [75]. Areas with maximum significant wave heights greater than 1.5 m are considered poor locations, as this height is the maximum height that the most sensitive vessels are unable to mitigate (tug boats and crew transfer vessels; see Ramachandran et al., 2022 [75] Table 3). Marginal locations are areas with maximum significant wave heights between 1 and 1.5 m, as these locations are manageable for some of the sensitive vessels in Table 3 of Ramachandran et al., 2022 [75]. The remaining areas are considered good locations for placement. Poor, marginal, and good locations for wave heights are shown in Figure 21 subplot (a).

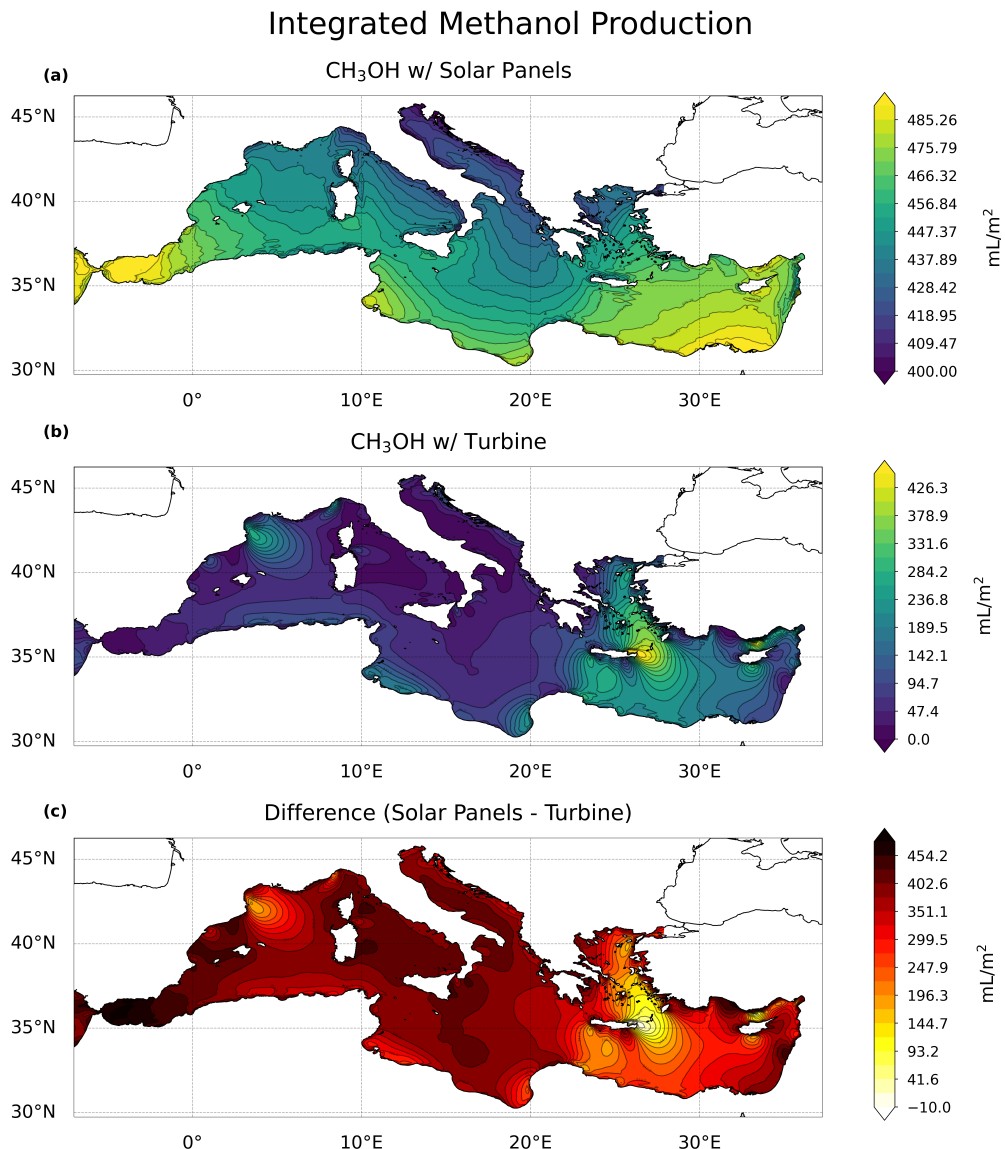

**Figure 20.** The integrated methanol production over the course of the ensemble averaged year for a simulated island running on solar power (**a**)), wind power (**b**)), and the difference between the two production simulations (**c**)).

For the results from Section 3.3, the thresholds to separate the production rates into the different categories are more arbitrary. However, optimal locations follow the trends that were highlighted in that section. Figure 21b shows the optimal locations for solar powered islands, in conjunction with the wave height information, with the poor locations featuring rates of less than 400 mL m$^{-2}$ integrated over the year. Marginal locations are marked by rates between 400 and 475 mL m$^{-2}$, with good locations marked by rates higher than 475 mL m$^{-2}$. Therefore, when considering both the limitations due to maximum significant wave heights and methanol production, the most optimal locations for solar powered islands are in the Alboran and Levantine Sea, with some locations along the northern coast of Africa.

For wind powered islands, poor locations are marked by areas producing less than 140 mL m$^{-2}$ over the course of a year. Marginal locations are marked by rates between 140 and 350 mL m$^{-2}$, with any location over 350 mL m$^{-2}$ marked as a good location. Figure 21c shows the optimal locations for wind powered islands, again in conjunction with the wave height limitations found in subplot (a). Wind powered islands have less optimal locations

than solar powered islands, with the best locations around the eastern side of the Greek island of Crete, and just north of the island nation of Cyprus.

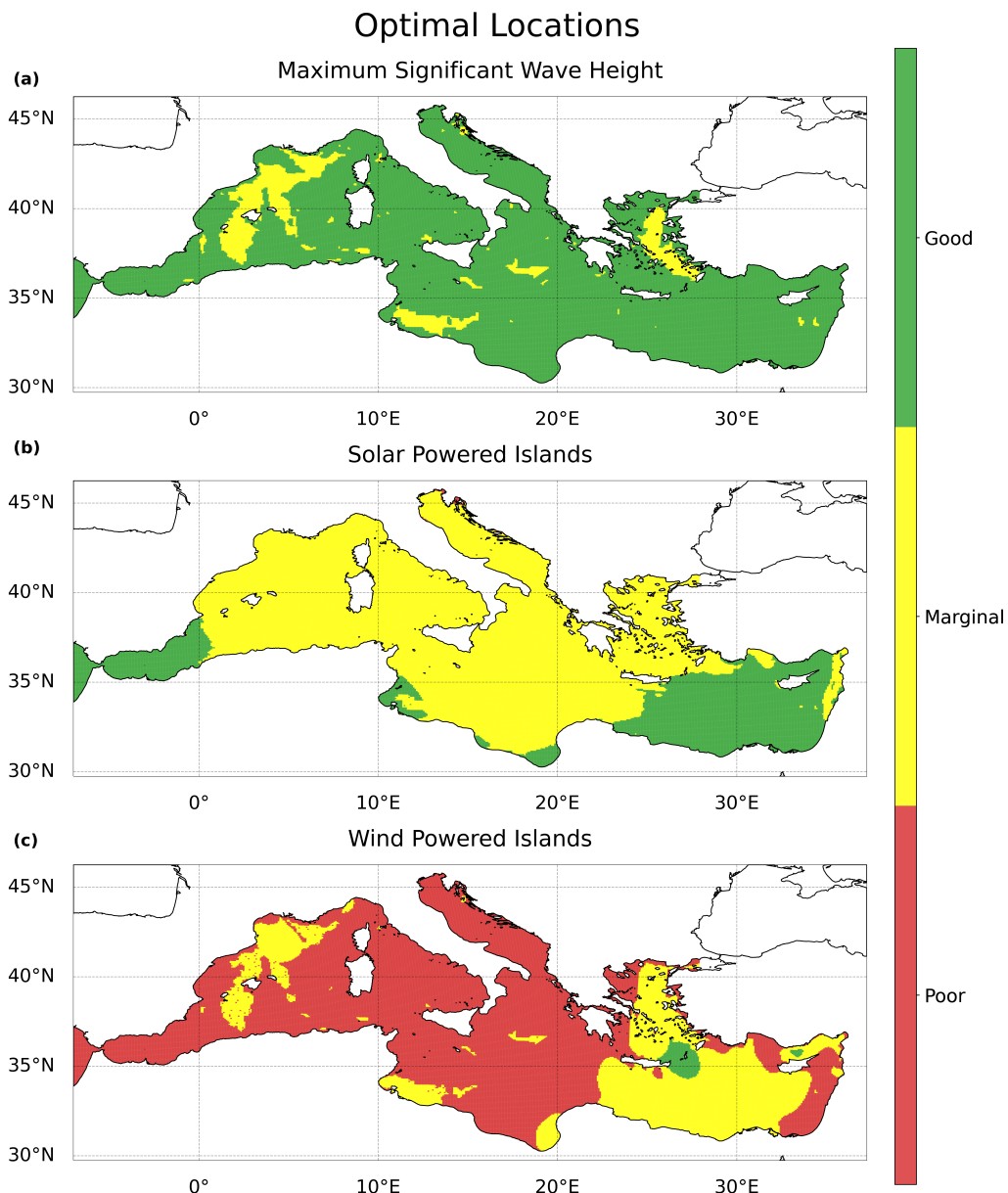

**Figure 21.** Optimal locations for solar or wind powered methanol producing islands. Subplot (**a**) shows the poor (none shown), marginal, and good locations according to the maximum significant wave heights (poor: above 1.5 m, marginal: between 1 and 1.5 m, good: below 1 m). (**b**) shows the combined above maximum significant wave height and solar powered methanol production of optimal locations (poor: below 400 mL m$^{-2}$ integrated methanol production over a year, marginal: between 400 and 470 mL m$^{-2}$, good: above 470 mL m$^{-2}$). (**c**) is the same as (**b**) but for wind powered islands (poor: below 140 mL m$^{-2}$, marginal: between 140 and 350 mL m$^{-2}$, good: above 350 mL m$^{-2}$).

### 3.5. Remote and Island Communities

Remote and island communities typically depend on oil for their energy needs [76]. The oil has to be shipped in, increasing the cost relative to the prices available to mainland communities, especially for islands (Smart Islands Initiative, https://www.smartislandsinitiative.eu/en/index.php; last accessed 13 June 2022). This

makes island communities prime candidates for the device described in this study, as they can produce methanol or methanol-derived products on site. The Mediterranean Sea contains a number of island communities, from the islands of Greece, Italy, and Spain, just to name a few. We will take two examples into consideration, the Greek island of Crete and the Spanish Balearic Islands.

For the first example, Crete consumed 3.95 TWh of gasoline and diesel during the year of 2016 [77]. Taking the energy density of methanol as $1.56 \times 10^7$ J L$^{-1}$, 0.91 GL of methanol would be needed to replace the consumed energy from oil. To produce this amount of methanol per year with a solar-powered methanol producing island, about 1962 km$^2$ of solar panels, or about a quarter of the surface area of Crete (8336 km$^2$), would be needed to supply the necessary power. If a wind powered device was used, 176447 SWT-3.6-120 turbines would be needed. The London Array construction cost was £1.8 billion for 175 turbines. If the cost is proportionally scaled, to supply Crete with the necessary methanol through wind power, the turbines would cost £1.8 trillion. Without considering the sea surface usage of either power method, this would not be a plausible solution for Crete. However, as part of a diversified energy mixture [76], storing energy as methanol and burning it when needed could be an effective alternative to hydrogen, as it is a much more stable substance to store.

For the second example, the entire economy of the Balearic Islands consumed 30.84 TWh of energy for the year of 2012 [78]. To supply these islands with with the same amount of energy in methanol, 7.12 GL of methanol would be needed. For a solar powered device, 15,863 km$^2$ of solar panels would be needed. For a wind powered device, 605,096,054 turbines would be needed. The energy consumption is an order of magnitude larger than Crete's oil consumption, with the required amount of power sources following suit. The same conclusions for Crete apply here as well.

## 4. Conclusions

The simulated methanol island presented here requires information from the ocean and atmosphere. When calculating the power requirement to run the device, the desalination process relies on the sea surface salinity and temperature, however, it is the only process to rely on the ocean surface variables. This is because the other potentially effected process, the degassing subprocess, is decoupled from the concentration of $CO_2$ in the surface waters. Eisaman et al., 2012 [29] used artificial seawater, as do other groups studying $CO_2$ extraction from seawater (3 of the 4 groups in Table 1 of Sharifian et al., 2021 [12] use artificial seawater), which has a constant concentration of $CO_2$, and therefore a varied concentration cannot be tested. This means it is difficult to accurately estimate the $CO_2$ given a certain concentration of dissolved inorganic carbon (DIC) in the feed flow. However, in the Mediterranean surface waters, the values of DIC do not vary too much from one side of the sea to the other, and are all around 2300 μmol kg$^{-1}$ [79]. Because of this, the spatial variability of DIC should not affect our results much, even if the extraction process was dependent on DIC concentration. The remaining two subprocesses, the electrolysis and reactor subprocesses, do not depend on the ocean variables nor atmospheric variables, at least not directly. Depending on the environment the methanol island is placed in, heating/cooling from the atmosphere or ocean on either subprocess could affect its performance; however, this can be addressed in the physical design of the island to minimize such influences. The result is, from the process side, that the system is relatively agnostic of the ocean surface variables, as the desalination process, the only process affected by the ocean, has a minimal contribution to the overall power requirement of the device, hence why the simulated methanol production efficiency is more or less constant over the entire basin.

Here, we only consider the surface waters of the Mediterranean, as these are the easily accessible waters to a methanol island. Deeper waters can be accessed if the reservoir of DIC in the surface layer is depleted, through siphons pulling inflow from the deeper layers. However, a parcel of surface layer water with a cross-sectional area of 1 m$^2$ and a depth of 25 m (average thickness of the mixed layer for the Med. Sea), contains roughly 2.6 kg of

carbon (if it is all extracted as $CO_2$ and assuming a DIC concentration of 2300 µmol $kg^{-1}$). At the flow rate of 10 L $s^{-1}$ used to simulate the island in this study, the 2.6 kg would be extracted in about 1.5 h. However, a surface current of 1 mm $s^{-1}$ would advect the water parcel a meter away in 0.5 h, preventing the device from extracting all of the DIC. If higher flow rates are used, a more in depth study of the DIC extraction, and ocean buffer system interaction with the atmosphere, will be needed to conclude the limitations of the resource replenishment. The state-of-the art research suggests that the replenishing rate of the DIC in the surface layers from the atmosphere may be the limiting factor for effective carbon capture methods utilizing the ocean [80].

The factor dominating the optimal locations for placing the methanol island is the power availability, due to the aforementioned spatially constant methanol production efficiency. As previously stated, the production efficiency behaves as a constant coefficient, such that the production per area of power generation varies the same as the power availability. This essentially merges our power availability constraint and methanol production constraint into a single production per area of power generation constraint that strongly depends on the power availability. Therefore, the atmospheric variables, the wind speed and solar forcing, provide the determining factor for methanol island placement, as they drive the power availability. The resulting optimal locations become apparent after integrating the production over the course of a year, with the Alboran, Levantine, and Cretan Sea being the best locations. These areas also escape the worst locations in terms of the maximum wave height (and by proxy significant wave height; even though the Cretan Sea is part of the Aegean Sea, the portion of the sea with increased levels of wind power generation avoid the larger maximum wave heights just north of it), and are therefore recommended for methanol island placement (refer to Figure 21).

Aside from the optimal locations for a methanol island, communities that could greatly benefit from this type of device include island communities. These communities require oil and gas but are required to ship these resources to the island, increasing the cost and carbon footprint of the imported resource. With a methanol island, island communities could produce their own fuel on location, avoiding the need to import oil and gas, gaining independence and safety (marine oil spillage) from outside resources. According to our results, the island of Crete could benefit from this type of system as part of a diversified energy economy, as well as being optimally located.

While we have estimated the environmental risk to the structure of a methanol island, we have not addressed the potential negative effects the operation of a methanol island could have on the local marine environment. Altering the ocean carbonate system will have an impact on marine biology. We have observed this through the effects of ocean acidification [81–86]. Increasing the acidity of the ocean negatively affects species with calcified bodies (pteropods, molluscs, crestaceans, etc.) [82,83,85], as these organism must put more energy into maintaining their calcified bodies. Conversely, the increased levels of dissolved $CO_2$ positively benefit phytoplankton, non-calcifying macroalgae, and seagrasses [83]. The metabolism of phytoplankton are $CO_2$ limited [81] and increasing their abundance will have knock on effects on species in the upper trophic levels. However, the negative effects of acidification on pteropods, molluscs, crestaceans, etc. will also cause knock on effects for species in the upper trophic levels, leading to an unreconciled situation where upper trophic species such as marine mammals and pelagic fish could be negatively and or positively effected by ocean acidification. In the Mediterranean, current research suggests pelagic fish experience more negative effects due to the rising sea temperatures rather than increasing acidification [85].

The removal of $CO_2$ from the ocean through the operation of methanol islands could reduce the impacts of ocean acidification on the marine biology by shifting the carbonate system to be more basic. However, to the authors' knowledge, there are no studies that attempt to investigate this. Ideally, the reversal of acidification would reverse its effects on marine organisms, positively affecting calcified organisms and negatively affecting the $CO_2$ limited growth of organisms forming the base of the marine trophic structure. However,

most likely there exists forms of biological hysteresis, in addition to other irreversible effects, that shift the system away from an ideal recovery. At this point in time, investigations on the effects of CDR on the environment have been limited to global and regional scale dynamics between the macro processes of the ocean, land, and atmosphere [87,88]. On these scales, an increasing ocean carbon sink could lead to land surfaces becoming carbon sources, reducing the overall benefit of trying to reduce atmospheric carbon. Additional research more concretely investigating the effects of $CO_2$ removal, such as the depletion of aqueous $CO_2$ and its recovery on the local scale, is greatly needed.

**Author Contributions:** D.K.J. wrote the bulk of the manuscript and performed the bulk of the modelling and analysis. V.S. contributed to the **pyseafuel** model utilized in this study. P.D. and C.T. are both the Principal Investigators and conceptualized the work, as well as provided valuable feedback and guidance on the manuscript and data analysis. All authors have read and agreed to the published version of the manuscript.

**Funding:** This research has received support from the 3rd Programme d'Investissements d'Avenir [ANR-18-EUR-0006-02].

**Data Availability Statement:** The global atmospheric $CO_2$ data is available from the NOAA Global Monitoring Laboratory at the url: gml.noaa.gov/ccgg/trends/. The remaining data presented in this study are available on the E4C Datahub (https://www.e4c.ip-paris.fr/#/fr/datahub/project/10; last accessed 21 November 2022) or upon request.

**Acknowledgments:** This work was conducted in the framework of the Energy4Climate Interdisciplinary Center (E4C) of Institut Polytechnique de Paris and Ecole des Ponts ParisTech. It has received support from the 3rd Programme d'Investissements d'Avenir [ANR-18-EUR-0006-02]. The authors would like to thank Jean-Claude Dutay (LSCE-IPSL) for his comments.

**Conflicts of Interest:** The authors declare no conflict of interest. The funders had no role in the design of the study; in the collection, analyses, or interpretation of data; in the writing of the manuscript, or in the decision to publish the results.

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
