# Peer review of "Offshore CO2 Capture and Utilization Using Floating Wind/PV Systems: Site Assessment and Efficiency Analysis in the Mediterranean"

_energies, doi:10.3390/en15238873_

Round 1

Reviewer 1 Report

The idea of the paper is novel and the carbon dioxide capture aiming at the island is a very bright spot.

However, this paper uses a non-public software package (Python package pyseafuel) to simulate the relevant numerical content, lacking public verification, and whether the acquisition and calculation of the relevant simulation can be reproduced needs further clarification.

Some comments are as follows:

1. Abstract is too lengthy and has no key points. It needs to be rewritten. Some conceptual contents are unnecessary to be put in it.

2. Introduction is not detailed enough to fill the defects of the literature. It needs to classify the relevant literature. The introduction and comparison of the existing software packages also need to be further explained.

3. A special device, called a "solar metal island" needs to introduce the structure of the device, and give pictures and other references.

4. How are windpowerlib, pvlib, and pyseafuel of Python package composed? You can show some detailed formulas and pseudo code as well as the underlying logic.

5. The preconditions and assumptions of some models and calculation results need to be given.

7. The layout of the paper is not particularly good, and it needs to be adjusted according to the content.

Reviewer 2 Report

In this manuscript a device named “solar methanol island" has been simulated using python package pyseafuel in order to extract carbon dioxide from the ocean and convert it to methanol. The amount of solar and wind power in the different area in the Mediterranean ocean as well as the maximum height waves are estimated in this simulation, in order to investigate the best locations for the methanol island. The idea of the work is quite interesting, and it is related to the sustainable energies and CO2 capture subjects which are the important aspects of researches, these days. However, there are some issues which are needed to be addressed as below.

My comments are as follow:

My main concern regarding this work is the effect of CO2 extraction on the ocean’s ecosystem. As stated in the manuscript “the decrease in dissolved CO2 reduces its partial pressure in the ocean, leading to the atmospheric CO2 to equilibrate with the ocean and reinsert atmospheric CO2 into the waters”.

  creatures in the oceans are sensitive to environmental changes, especially on CO2 and PH levels. Extracting CO2 from oceans will change the partial pressure of CO2 in water. The question is how long is going to take to resolve CO2 of the atmosphere in the seawater and reaching to equilibrium again? Perhaps many years is required to reach to this new equilibrium and during this period the lifestyle of the various plants and animal would be changed. The CO2 extraction also increase the PH levels of seawater, resulting in dramatic physiological effects on many species.

Is there any research performed in this area which investigate the effect of CO2 extraction and changing in PH level of seawater on the ocean’s ecosystem? The authors are recommended to answer to this question and add their response in the manuscript.

In the introduction the authors stated: “When we simulate the production at these locations, a 10 L s-1 seawater inflow rate produced 494.21, 495.84, and 484.70 mL m-2 over the course of a year, respectively.” Do you mean produced 494.21, 495.84, and 484.70 mL m-2 methanol?  Please edit.

The authors are recommended to change “model” in keyboard section to “Simulation model”.

Line 326 in the manuscript: “H2 to CO2 mole ratio of 25.83”. The authors are recommended to change “mole ratio” to “mole percentage” or change 25.83 to 0.25 mole ratio.

Comments on the references:

Add page number to reference 10

Add the chapter number to reference 21

Are references 23 and 50, book or report? add more information.

The page number for some references are not completed. (The end page is not included in some references).

Round 2

Reviewer 1 Report

The quality of the revision has been improved and the current version is ready for publication.